# Directed attenuation to enhance vaccine immunity

**Rustom Antia** [1]*, **Hasan Ahmed**[1], **James J. Bull**[2]

**1** Department of Biology, Emory University, Atlanta, Georgia, United States of America, **2** Department of Biological Sciences, University of Idaho, Moscow, Idaho, United States of America

* rantia@emory.edu

**Data Availability Statement:** All relevant data are within the manuscript and its Supporting information files.

**Funding:** RA and HA acknowledge support from the National Institutes of Health (U01 AI150747, U01HL139483 and U01AI144616). JB

## Abstract

Many viral infections can be prevented by immunizing with live, attenuated vaccines. Early methods of attenuation were hit-and-miss, now much improved by genetic engineering. However, even current methods operate on the principle of genetic harm, reducing the virus's ability to grow. Reduced viral growth has the undesired side-effect of reducing the host immune response below that of infection with wild-type. Might some methods of attenuation instead lead to an increased immune response? We use mathematical models of the dynamics of virus with innate and adaptive immunity to explore the tradeoff between attenuation of virus pathology and immunity. We find that modification of some virus immune-evasion pathways can indeed reduce pathology yet enhance immunity. Thus, attenuated vaccines can, in principle, be directed to be safe yet create better immunity than is elicited by the wild-type virus.

## Author summary

Live attenuated virus vaccines are among the most effective interventions to combat viral infections. Historically, the mechanism of attenuation has involved genetically reducing the viral growth rate, often achieved by adapting the virus to grow in a novel condition. More recent attenuation methods use genetic engineering but also are thought to impair viral growth rate. These classical attenuations typically result in a tradeoff whereby attenuation depresses the within-host viral load and pathology (which is beneficial to vaccine design), but reduces immunity (which is not beneficial). We use models to explore ways of directing the attenuation of a virus to avoid this tradeoff. We show that *directed attenuation* by interfering with (some) viral immune-evasion pathways can yield a mild infection but elicit higher levels of immunity than of the wild-type virus.

## Introduction

Many highly successful viral vaccines use live attenuated viruses, which are typically variants of the wild-type virus that have a genetically reduced capacity to grow [1–3]. Attenuated vaccines are considered to offer superior protection over other types of vaccines [4]. Early

acknowledges support from National Institutes of
Health (GM122079). The funders had no role in
study design, data collection and analysis, decision
to publish, or preparation of the manuscript.

**Competing interests:** The authors have declared
that no competing interests exist.

methods of attenuation were haphazard, relying on adaptation to viral growth in unnatural,
artificial conditions to indirectly evolve a virus with reduced capacity to grow in the natural
host. The fact that attenuation by this largely blind method was repeatable at all suggests
that there is a broad window of viral growth reduction compatible with practical vaccine
attenuation.

Viral attenuation entered a new era with genetic engineering, when it became possible to
achieve quantitative reductions in viral growth rate and to block evolutionary reversion of
attenuation [5–10]. Much of the guesswork in reducing viral growth rate is eliminated with
genetic engineering approaches, and the engineering can likewise slow or block the vaccine's
ability to reverse the changes and become virulent. Now, genetic engineering may be ushering
in a third era, one in which attenuation is designed to enhance the immune response. Such
'directed attenuation' will require genetically altering (engineering) the virus in a manner that
reduces pathology while enhancing the level of protective immunity. In this paper we use the
term directed attenuation to refer to engineering vaccine genomes to influence the level of
immunity.

The need for new methods of viral attenuation is not necessarily obvious. Many attenuated
vaccines in use—created by old methods—have brought about dramatic reductions in the
prevalence of infections with few side effects, so is there anything to be gained from changing
their design? However, live attenuated vaccines typically do not elicit as much immunity as
natural infection [11–16]. This can potentially result in reinfection with circulating virus (e.g.
for mumps where the loss of immunity necessitates periodic boosting [12, 17]). Furthermore
at a more fundamental level, vaccines remain elusive for many viruses [18–22] and require
exploration of alternative approaches to vaccine design.

It is not a trivial matter to engineer a live vaccine and knowingly vary the immune response.
It has become practical to engineer a virus to knowingly reduce the pathology (i.e., to attenu-
ate, [4]). However current attenuation compromises the growth of the virus, leading to a
reduced viral load that typically elicits a smaller immune response [23]. Ideally, it would be
better to attenuate the virus without compromising long term immunity, perhaps even elevat-
ing it. Doing so requires understanding the complex interactions and feedbacks between virus
growth and generation of adaptive immunity.

The first step in directing attenuation involves characterizing the different viral proteins
and how they interact with the cells and molecules involved in the generation of immune
responses—the subject of recent advances in molecular and cellular virology and immunology
(reviewed in [24, 25]). Yet a qualitative understanding of these pathways is not enough: non-
linear feedbacks between virus growth and immunity make it hard to comprehend the effect
of modifying viral genes and proteins on the level of immunity generated. Is it feasible to
enhance immunity without increasing pathology? Targeting pathways that interfere with host
immunity may simply result in more rapid virus clearance (less pathology) and consequently
less clonal expansion and immunity at the end of the infection.

The approach to directed attenuation presented here is to develop a mathematical model
that incorporates key elements of our current understanding of the replicating virus and the
immune response it elicits. We then use the model to explore whether directed attenuation
might simultaneously reduce pathology and enhance immunity, and if so, to identify candidate
pathways for virus engineering.

## Basic model

We use a deliberately simple model for the dynamics of infection with a virus—whether wild-
type or vaccine. Let $V$ equal the virus density and $Z$ and $X$ equal the magnitude of the innate

and adaptive responses respectively; all are functions of time.

$$
\begin{aligned}
\text{(virus)} \quad \frac{dV}{dt} &= rV - k_Z ZV - k_X XV \\
\text{(innate immunity)} \quad \frac{dZ}{dt} &= s_Z(1-Z)\frac{V}{\phi_V + V} - d_Z Z \\
\text{(adaptive immunity)} \quad \frac{dX}{dt} &= s_X X \frac{Z}{\phi_Z + Z} \\
\text{(pathology)} \quad P &= \max(V)
\end{aligned}
\tag{1}
$$

The virus grows exponentially at rate $r$ in the absence of innate and adaptive immunity. Virus is killed by innate and adaptive immunity at rates proportional to current immunity levels and the parameters $k_Z$ and $k_X$ respectively. Innate immunity, $Z$, is activated by infection and saturates at 1, so $Z$ equals the fraction of the maximum possible value of innate immunity. Adaptive immunity, $X$, grows in proportion to innate immunity ($Z$). This representation is a modification of that usually used, where the stimulation of adaptive immunity depends only on the amount of virus $V$. It is justified on biological grounds that the generation of an adaptive immune response requires stimulation by activated innate immunity (e.g. a danger signal); the observation that virus antigen is typically presented to antigen-specific immune cells in the lymph nodes for an extended period, potentially after live virus is cleared, also supports disconnecting adaptive immunity expansion from live virus [26–29]. We let the extent of pathology, $P$, equal the maximum level of virus. The basic form is similar to that used previously [30–35]. Table 1 shows the parameter ranges used, consistent with the known characteristics of immune responses to acute infections [36]. In all simulations, we rescale the initial virus and adaptive immunity to 1. The rates of viral growth and immunity ($r$ and $s$) are consistent with that observed from infection dynamics in mouse model systems [37]. The initial virus density is typically small, so virus proliferation is required for stimulation of innate immunity ($\phi_V \gg 1$); innate immunity decays with a time scale of days ($d_Z \sim 1$). The rate constant for control of virus by innate immunity ($k_Z$) is chosen to allow it to control virus replication in a few days; that of adaptive immunity is chosen to be small because considerable clonal expansion is needed for adaptive immunity to reach a sufficiently high density to clear the infection ($k_X \ll 1$).

**Table 1. Parameters of the model.**

| Parameter | Description | Value and range |
|:---:|:---|:---:|
| $r$ | virus growth rate | $3\mathrm{d}^{-1}$ $(2-4)$ |
| | Innate immunity | |
| $s_Z$ | rate of activation of innate immunity | $.1\mathrm{d}^{-1}$ $(.06-.14)$ |
| $\phi_V$ | sensitivity of innate immunity | $10^2\mathrm{v}(10^1 - 10^3)$ |
| $d_Z$ | waning of innate immunity | $1\mathrm{d}^{-1}$ $(.4-1.6)$ |
| $k_Z$ | killing of virus by innate immunity | $30(\mathrm{zd})^{-1}$ $(10-100)$ |
| | Adaptive immunity | |
| $s_X$ | rate of growth of adaptive immunity | $.1\mathrm{d}^{-1}$ $(.5-1.5)$ |
| $\phi_Z$ | sensitivity of adaptive immunity | $.02\mathrm{z}(.002-.2)$ |
| $k_X$ | killing of virus by adaptive immunity | $10^{-2}(\mathrm{xd})^{-1}$ $(10^{-3} - 10^{-1})$ |

Parameters of the model and the values and ranges used in simulations with units for measurement (d = days, v = virus, x = adaptive immunity, z = innate immunity).

The model is intended to capture basic properties of the immune response, helping discover unintuitive interactions and feedback effects that arise as a consequence of viral growth that interferes with the host's immune response. This model does not pretend to capture all the known complexity of mammalian immune systems. For example, it does not include different populations of innate and immune cells or cytokines generated during immune responses or antigen-independent proliferation of adaptive immunity after initial activation [38–40]. The inclusion of these details at this stage would lead to a complex model with no hope of parameterization.

## Results

### Dynamics of acute infection

Dynamics of an acute infection are straightforward (Fig 1, showing changes in viral titer, innate and adaptive immunity over the course of infection for one set of parameter values). The virus initially grows exponentially. Soon thereafter the innate immune system is triggered and begins to limit viral growth. The adaptive immune response is delayed relative to innate immunity, it is responsible for clearance of the infection and is subsequently maintained over the time scale considered. Innate immunity is temporary and decays to zero soon after the virus is cleared.

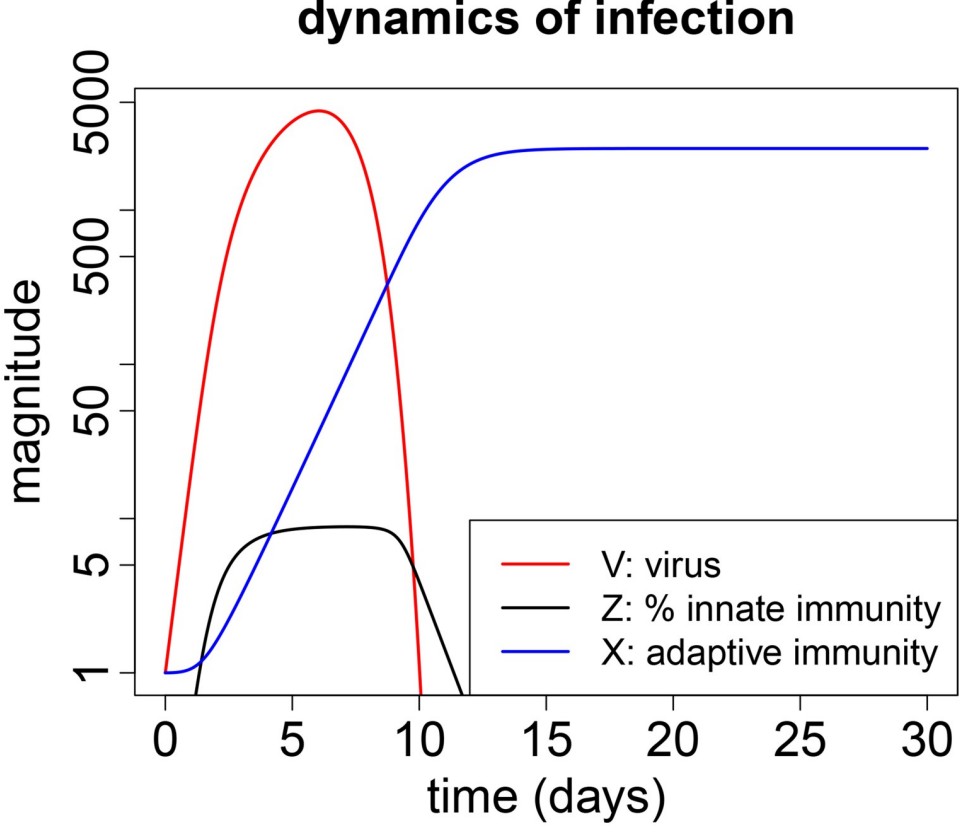

**Fig 1. Typical dynamics of an acute infection.** Virus ($V$) is shown in red, innate immunity ($Z$) in black and adaptive immunity ($X$) in blue. The scale for virus and adaptive immunity is fold change over the initial value ($V(0)$ and $X(0)$ are set to 1). The scale for innate immunity is percent of its maximum possible value (100), and in this simulation $Z$ attains only about 10% of its maximum. Parameters are chosen for a biologically relevant regime as described earlier [36] and are shown in Table 1.

## Classic attenuation by reducing virus growth

The original approach for attenuating a virus was to grow it in unnatural conditions (e.g., novel host or cell culture). Adaptation to the unnatural conditions often—but unpredictably—reduced growth rate in the primary host (humans), with consequent loss of pathogenesis. Newer methods of engineered attenuation also lower viral growth rate but do so far more predictably [4, 10]. With a viral growth rate less than wild-type, the attenuated virus attains a smaller peak density before clearance and thus elicits less adaptive immunity. To wit, a single infection with wild-type measles, mumps or rubella viruses typically induces lifelong immunity [41, 42], while vaccine-induced immunity frequently requires boosting, as indicated by the CDC immunization schedule [17].

This basic trade-off provides a baseline that should be reproduced by any reasonable model of viral-immune dynamics: reduced viral growth rate should result in reduced viral density before clearance (reduced pathogenesis). There should be a consequent reduced stimulation of adaptive immunity and a lower final level of adaptive immunity. Our model indeed generates the expected patterns (Fig 2). This pattern highlights the fundamental question of our study: is it possible to engineer an attenuation that is better than achieved by merely reducing viral growth rate—can we arrange pathogenesis to go down but immunity go up? Our approach to this question involves varying virus-affected parameters and observing changes in pathogenesis and in immunity, as done next.

## Attenuation by suppressing virus immune evasion pathways

Viruses have pathways that interfere with the innate or adaptive immune responses [43–47]. As these reduce the magnitude or effect of immunity, they are candidate pathways, that if targeted, could lead to an increase in immunity. We model the suppression of these viral pathways as changes in the parameters that correspond to attenuation strategies—reducing pathogenesis (Table 2). Note that suppression of a viral pathway may result in an increase of the parameter value—which can occur when the wild-type virus depresses a host anti-viral response.

Virus strategies to evade innate immunity are modeled as low values of $s_Z$ (the maximum rate of stimulation), high values of $\phi_V$ (the level of virus needed for stimulation of innate immunity) and high values of $d_Z$ (the decay rate of innate immunity). Similarly, viruses avoid

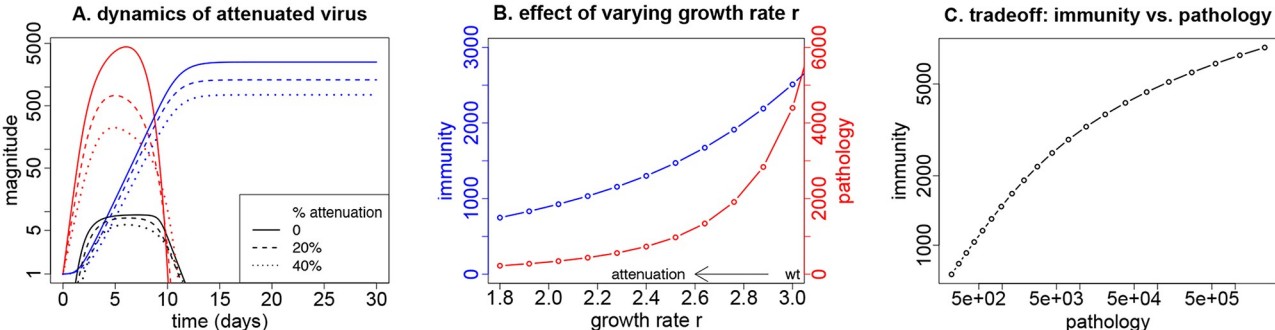

**Fig 2. Attenuation by reducing the growth rate $r$ of the virus.** Solid lines indicate wild-type; dashed and dotted indicate attenuated. Reducing virus growth rate results in lower viral load as well as a reduction in the final level of adaptive immunity. (A) Dynamics for the wild-type infection in solid lines (red for virus, blue and black for adaptive and innate immunity) and for viruses with a 20% (dashed) and 40% reduced growth rate (dotted). (B) Impact of the degree of attenuation (reduction in $r$) on both the final level of adaptive immunity (blue) and the pathology (maximum virus load, red). (C) The tradeoff between pathology and peak adaptive immunity from changing growth rate $r$: reducing the growth rate results in lower pathology but also lower immunity. Parameters values are given in Table 1.

**Table 2. Candidates for directed attenuation.**

| Evasion pathway targeted (parameter & direction of change for vaccine) | Vaccine outcomes | |
|---|---|---|
| | pathology | immunity |
| _Innate immunity_ | | |
| max. activation rate for innate immunity ($s_Z \uparrow$) | decrease | decrease |
| sensitivity of innate immunity to antigen ($\phi_V \downarrow$) | decrease | no change |
| **decay of innate immunity ($d_Z \downarrow$)** | **decrease** | **increase** |
| killing by innate immunity ($k_Z \uparrow$) | decrease | decrease |
| _Adaptive immunity_ | | |
| **max. rate of clonal expansion ($s_X \uparrow$)** | **decrease** | **increase** |
| **sensitivity of immunity to antigen ($\phi_Z \downarrow$)** | **decrease** | **increase** |
| killing by adaptive immunity ($k_X \uparrow$) | decrease | decrease |

Summary of candidates for directed attenuation arising from changes in a single parameter. Desired outcomes that result in a decrease in pathology and an increase in immunity are shown in boldface.

stimulation of adaptive immunity by low values of $s_X$ (the maximum rate of proliferation of immune cells) and high values of $\phi_Z$ (the level of stimulation required to induce proliferation of adaptive immune cells). Viruses can also limit killing by the innate and adaptive immunity with low values of the rate constants $k_Z$ and $k_X$ respectively. Attenuation thus involves changing those parameters in the opposite directions, reducing pathology. But our interest also lies in which of these changes will have the additional effect of increasing the final level of adaptive immunity, or at least not lowering it.

**Blocking immune-evasion genes attenuates the infection; some changes can also enhance immunity.** Blocking (reducing) any viral immune-evasion pathway leads to attenuation of the virus (Fig 3). All are thus potential routes for generation of a live attenuated virus vaccine. But how might those changes affect the level of immunity? Blocking pathways that make the virus more sensitive to clearance by either innate ($k_Z$) or adaptive ($k_X$) immunity results in more rapid control of the virus but a lower final level of immunity. In contrast, blocking pathways that slow the generation of adaptive immunity (either by changing the immune growth rate, $s_X$, or sensitivity of adaptive immunity, $\phi_Z$) results in greater stimulation and a _higher_ final level of adaptive immunity. Finally, suppressing the viral pathway accelerating innate immune decay ($d_Z$) prolongs the stimulation of adaptive immunity and increases its final level.

Thus attenuation by modifying immune-evasion pathways described by parameters $s_X$, $\phi_Z$ and $d_Z$ generates the hoped-for outcome of reduced pathology with increased immunity. Attenuation of other immune-evasion pathways, such as those associated with changes in the stimulation of innate immunity ($s_Z$ or $\phi_V$) or susceptibility to killing by innate or adaptive immunity ($k_Z$ and $k_X$), does not result in higher immunity. A more detailed consideration of the effect of each parameter on the dynamics of infection and immunity is considered in S1 Text.

**Pathology-immunity plot enables comparison of attenuation strategies.** The effects of different attenuation pathways may be directly compared by plotting them together in a grid of pathology and immunity (Fig 4). The ideal live attenuated virus vaccine would generate lower pathology but higher immunity than infection with the wild-type virus, corresponding to the upper left quadrant (where wild-type is taken as the center). It is straightforward to see that directed attenuation works for the same three parameters identified above, $d_Z$, $s_X$ and $\phi_Z$. It is also seen that increasing sensitivity of innate immunity (decreasing $\phi_V$) leads to

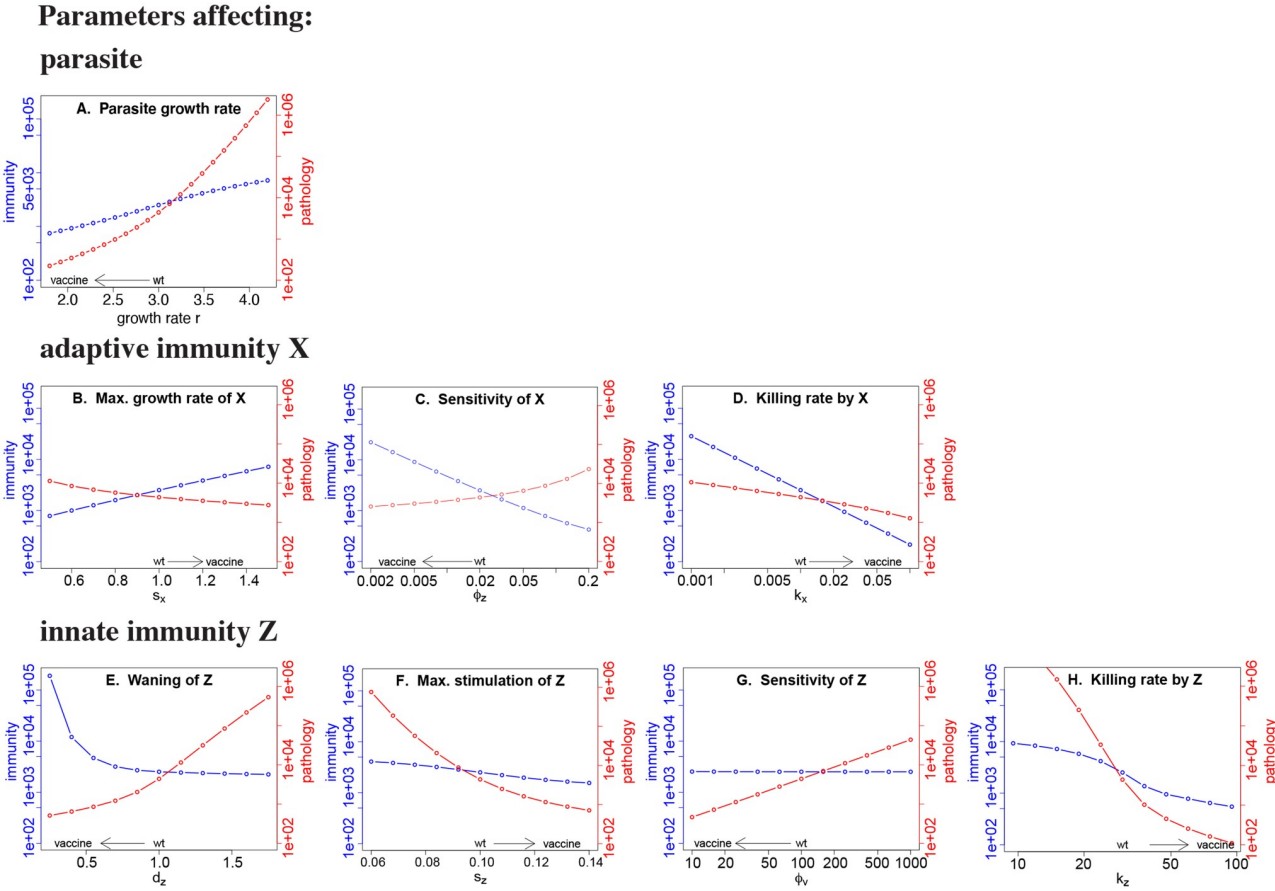

**Fig 3. Effect of changing single parameters on the levels of pathology and immunity.** Red curves give the pathology, with scale on the right vertical axis; blue lines give the final level of adaptive immunity, with scale on the left vertical axis. The baseline parameters chosen for the wild-type virus are indicated by 'wt' on the horizontal axis. A vaccine strain would be designed to lower pathology, and the arrow immediately above the horizontal axis gives the direction of change in the parameter value that would reduce pathology. The goal of directed attenuation is to achieve a decline in the red curve and an increase (or no change) in the blue curve relative to wild-type; several attenuation designs achieve this outcome. Baseline parameters are as in Fig 1 except for the parameter whose value is changed in the panel.

attenuation of pathology without compromising immunity. A summary of the effect of different directed attenuation strategies by single parameter changes is included in Table 2 (above).

## Combining attenuation strategies

Two potential problems with directed attenuation of a single viral anti-immune pathway are (i) reversion of attenuation during growth in the patient, and (ii) potential harm when vaccinating immunocompromised individuals. Given modern methods for engineering attenuation, the first of these (reversion) may not be a significant problem; we thus focus on inadvertent vaccination of immunocompromised individuals. The problem is especially apparent if a virus is attenuated by deletion of an antiviral strategy that targets a defense pathway lacking in the patient, in which case there is no difference between infection by the wild-type virus and infection by the vaccine. The conventional method of attenuation—which results in a reduction of the growth rate, *r*—avoids this problem, or at least makes the infection less severe than that with wild-type virus. Overcoming this problem with directed attenuation may require a combination strategy. One type of combination is to block immune evasion while

## tradeoff between pathology and immunity

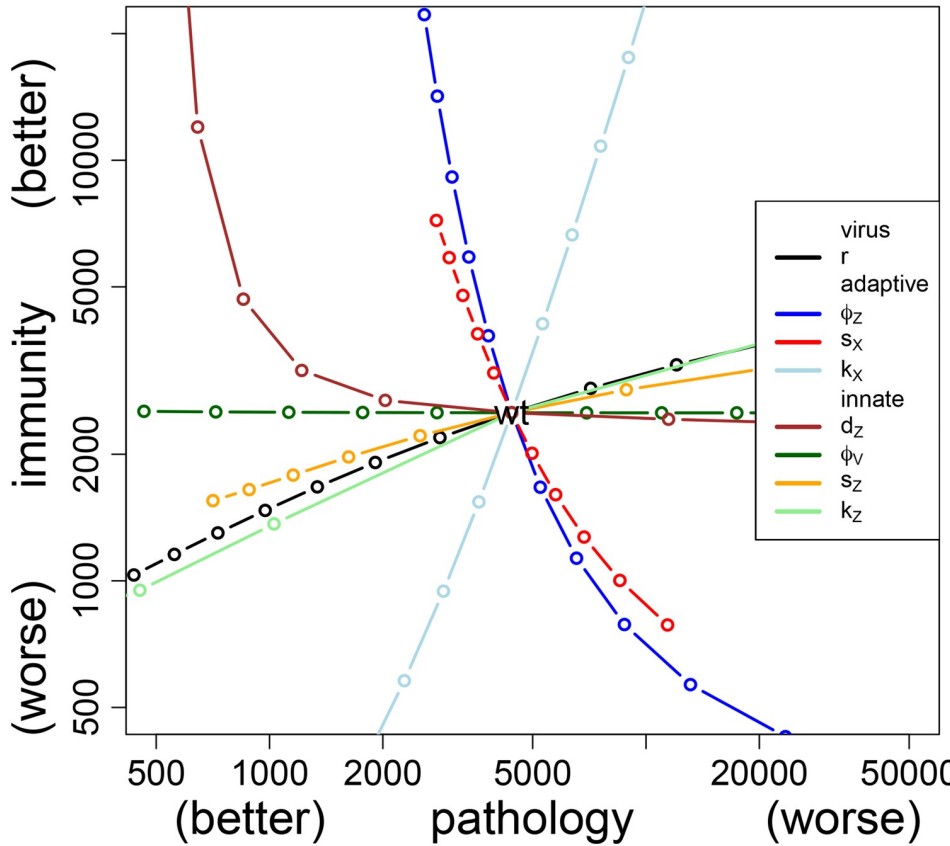

**Fig 4. Comparison of immunity-pathology tradeoffs generated by changing single parameters.** Wild-type values are given at the intersection of the curves, so viable attenuation strategies would lie to the left. As the goal is to attenuate and to increase the immune response, the desirable attenuation strategies lie in the upper left quadrant. The tradeoff for the classic mode of attenuation—lowering growth rate $r$ (black line)—has the undesirable effect of lowering immunity, a pattern mimicked by changes in several other parameters. In contrast, decreasing the rate of loss of innate immunity ($d_Z\downarrow$ (brown)), or increasing the rate of proliferation or sensitivity of adaptive immunity ($s_X\uparrow$ (red), $\phi_Z\downarrow$ (blue)) leads to lower pathology and increased immunity.

separately reducing growth rate. Another type of combination is to block both evasion of adaptive immunity and evasion of innate immunity.

A worry with combination strategies is that the reduced immunity might be too severe: can wild-type immunity be attained in a virus with classic growth rate reduction when an anti-immunity defense is also disabled? The answer appears to be affirmative for some combinations (Fig 5). The top four panels show the combined effect of reducing viral growth rate (going from right to left on the horizontal axis) together with changing one immune parameter (on the vertical axis). Suppressing an immune-evasion pathway that changes the sensitivity of the adaptive immunity (reduces $\phi_Z$) or the decay rate innate immunity (i.e. increases $d_Z$) restores vaccine immunity to the level elicited by the wild-type virus (right column), and this is done without compromising the level of attenuation of pathogenesis that was obtained by the reduction in $r$ (left column).

Combinations are possible without altering growth rate ($r$), and some combinations may likewise result in reduced pathology with enhanced immunity (Fig 5). The combination

## Effect of changing growth rate and other parameters on:

**Fig 5. Heat maps for the $\log_{10}$ fold changes in pathology (top row) and immunity (bottom row) for changes in two parameters.** The effect of changing a single parameter alone is seen by moving parallel to the respective axis. The goal of attenuation is to reduce pathology from wild-type values (increase the level of blue, top row) and to increase immunity (increase redness, bottom row). Values of wild-type virus are given in upper right of each panel, values of the prospective vaccine in lower left. The goal is to have pathology become increasingly blue and immunity become increasingly red in traversing from wild-type to vaccine. The conventional attenuation strategy arising from reductions in the parameter $r$ is seen to reduce both pathology and immunity (moving left along the horizontal axis in any of the left four panels). However, combining reductions in $r$ with increasing the sensitivity of the adaptive immune response (i.e. decreasing $\phi_Z$) or increasing the duration of innate immunity (i.e. increasing $d_Z$) restores the level of immunity generated by the vaccine to that induced by the wild-type virus while reducing pathology (left two columns). The right column shows attenuation achieved by changes in a pair of parameters that does not include $r$.

shown that does not change growth rate (bottom row) suppresses viral evasion of innate immunity (increasing $\phi_V$) while sensitizing adaptive immunity to stimulation by innate immunity (increasing $\phi_Z$).

One of the striking observations from Fig 5 is that the effect of varying parameters is substantially different for the 3 cases. In the plots in the first column, immunity changes in response to both parameters, but pathology changes largely in response to the growth rate $r$. In the plots in the last column, immunity is strongly affected by the sensitivity of the adaptive response $\phi_Z$ but largely unaffected by the sensitivity of the innate immune response $\phi_V$, whereas pathology shows much the opposite pattern. The middle column shows both pathology and immunity being affected by the combination of parameter values. These results reinforce the unintuitive nature of directed attenuation, and illustrate how models can be useful tools to understand the consequences of the non-linearities and feedbacks for different vaccination strategies.

### Robustness of directed attenuation to model formulation

The presentation above used a single model of immunity and viral dynamics. To address the possibility that directed attenuation is an outcome specific to that model, two other models

were studied (see S1 Text). Directed attenuation was found to be attainable in those as well. The two models differed from the model above as follows. In one model, the stimulation of the adaptive immune response was dependent only on the amount of virus antigen. In a second model, the stimulation of adaptive immunity depended on both the activation of innate immunity and the amount of virus antigen. Although the details of how to achieve directed attenuation differed somewhat across these models, directed attenuation was possible for some parameter changes, commonly those involved in viral suppression of immunity. The fact that directed attenuation can be attained across different models suggests that it may be a general principle.

## Discussion

Attenuated vaccines have been the mainstay of viral vaccines for close to a century [2–4]. Attenuation is typically achieved by evolving the virus (for example by growth in a new environment) or genetically modifying it so as to reduce its growth rate following infection of the host [5–10]. Here we suggest an alternative approach to attenuation, one that directs the virus toward reduced viral defenses against host immunity. This approach to attenuation is suggested by the fact that many viruses directly interfere with the immune response [43–49], and those interference genes are obvious targets for genetic engineering. Using a simple computational model of the immune response to viral infection, we found that disruptions of viral anti-immune pathways invariably led to reduced pathology, whereas disruption of some, not all, of these pathways did so without compromising immunity, even increasing the level of adaptive immunity in some cases.

 Our model helps identify which virus immune evasion pathways might be disabled to achieve the desired outcome of increasing the level of immunity generated. We can intuit the consequences of deleting some of the immune evasion pathways. Deleting pathways that make the virus resistant to clearance by innate and adaptive immunity is modeled by increasing the rate of virus clearance by innate or adaptive immunity ($k_Z$ and $k_X$). This results in more rapid virus clearance and more rapid waning of innate immunity, and consequently a shorter duration of stimulation of adaptive immunity, and thus a lower final level of adaptive immunity. More promising targets for genetic engineering include virus genes that affect the generation of innate or adaptive immune response (model parameters $\phi_V$, $s_Z$, $\phi_Z$ and $s_X$). It is hard to intuit the consequences of changes in these parameters because changes in these parameters affect the final level of adaptive immunity in multiple ways. The first is the more rapid generation of adaptive immunity, either directly (for $s_X$ and $\phi_Z$) or indirectly via faster generation of innate immunity (for $s_Z$ and $\phi_V$). The second effect arises as a consequence of the faster generation of adaptive immunity. The faster generation of adaptive immunity leads to faster virus clearance which leads to earlier waning of innate immunity, and this curtails the duration of stimulation of adaptive immunity. These two effects work in opposite directions, the former resulting in an increase in the rate of generation of adaptive immunity and the latter in a decrease in the duration of expansion of adaptive immunity. Intuition is not enough and we thus require mathematical models to determine the consequences of combining these two effects. Finally, deleting virus pathways that increase the rate of inactivation of innate immunity (modeled by lowering $d_Z$) tend to have a beneficial effect, because it prolongs the duration of stimulation of the adaptive immune response. In the S1 Text we discuss the effects of changes in these parameters on the dynamics of virus and immunity in more detail.

### Viruses do evade immunity

Recent studies have discovered many pathways used by viruses to evade the innate and adaptive immune responses [43–47, 49]. Poxviruses are large DNA viruses with much of their

genome encoding immune-evasion pathways [45, 46, 50]. These pathways target host type 1 interferon, tumor necrosis factors and the complement pathway of innate immunity. They also evade adaptive immunity by downregulating antigen presentation and by blocking costi-mulatory pathways and the apoptotic response. Adenoviruses are medium-sized, non-envel-oped viruses containing a double stranded DNA genome. They encode immune-evasion pathways that inhibit tumor necrosis factor activity, and also evade adaptive immunity by downregulating antigen presentation [51–53].

One of the best studied immune-evasion genes encodes the NS1 protein of the influenza virus, which interferes with multiple stages of the type 1 interferon signaling cascade [54–58]. Mutant influenza viruses lacking the NS1 gene are highly attenuated in wild-type (interferon-competent) mice, but not in IFN-incompetant systems such as STAT 1 knockout mice [59]. Viruses with a truncation or deletion of their NS1 gene have been shown to be promising can-didates for a live attenuated vaccine in chicken [60, 61]. Type 1 interferon plays a role in reduc-ing virus replication, corresponding to a decrease in the growth rate of the virus ($r$) in our models, so we expect this to result in less pathology and a smaller immune response. This is consistent with the outcome of experimental infections of chicken with influenza virus vac-cines that have deletions in the NS1 protein [61]. We suggest that it might be worth exploring the possibility of targeting other virus immune-evasion pathways so that the immune response is increased.

An instructive example of virus engineering is that described by Jackson et al [62] who inserted a mouse IL-4 gene into ectromelia virus (mousepox). The engineered virus caused fatal infections of mice and even killed mice that were immune to the wild-type virus. This shows that: engineering virus immune evasion pathways is possible, that the result might have been predicted by relatively simple understanding of immunity (IL-4 activates the Th2 immune response and thereby suppresses the Th1 responses required for defense against viruses [63]), and that the outcome of engineering virus immune evasion pathways can result in viruses with increased virulence (indicating caution is needed).

## How to direct attenuation

A combination of experiments and modeling approaches would facilitate achieving the desired goal of engineering a virus vaccine with reduced pathology but that generates enhanced immu-nity. We certainly do not have sufficiently accurate quantitative models of the dynamics of infections and immunity to rely principally on the models. And a purely empirical approach has the problem of choosing between the large number of possible genes (and gene combina-tions) that could be the target of genetic manipulation. Intuiting the consequences of different mutations is challenging because of the non-linear feedbacks between control of virus and gen-eration of host immunity. Consequently, models that incorporate the relevant details of spe-cific virus immune-evasion pathways may help suggest combinations of pathways to be targeted, limiting the experimental effort needed, and analysis of the results will in turn allow refinement of the models.

At a basic engineering level, we suggest that in addition to compromising virus immune-evasion genes (e.g., knockouts or codon deoptimizations), directed attenuation can involve up-regulation or addition of genes that enhance the generation of immunity. Candidates for up-regulation or addition include pathogen associated molecular patterns (PAMPS) that induce the appropriate form of immunity. Studies on surrogates of protection [64] could help identify appropriate PAMPS for inclusion in the virus.

One consideration is the possibility of virus reversion. Gene additions are more likely to be prone to reversion through deletions and point mutations than gene deletions [65]. However,

Bull et al [65] suggested that within-host vaccine evolution can compromise vaccine immunity only when the extent of evolution during vaccine manufacture is severe, and that this evolution may be avoided or mitigated by suitable choice of the vaccine inoculum.

We have focused on virus pathways that compromise the magnitude of the immune response. Similar principles could be applied to virus pathways that skew the generation of immunity (e.g. the Th1 vs Th2 responses) away from responses that are most effective at clearing the virus. We could also consider pathways that compromise the maintenance of memory. Enhancing the longevity of immunological memory may have relevance for infections with viruses such as coronaviruses that may not induce lifelong immunity. It could also be extended to inactivated virus vaccines where genetic engineering to remove genes for virus proteins that inhibit the generation of immune responses may (if the proteins function in the absence of virus replication) enhance the immune response elicited by vaccination with the killed virus. Another extension would be to explicitly incorporate scenarios where the extent of attenuation depends on virus density, as has been suggested for some non-selfish virus evasion strategies [66]. Specific elaborations should be based on detailed consideration of a vaccination scenario for a specific virus. Our results can be used as a motivation for pursing directed attenuation but should not be used as a recipe for choosing the genes to attain directed attenuation.

**Summary.** This study uses models to explore the consequences of attenuating a virus by targeting the interaction of the virus with the host's immune response, contrasting with the more traditional approach of attenuation by reducing viral growth rate. This 'directed' approach allows us to explore how new vaccines may be designed with the goal of maximizing the level of protective immunity generated. Contrary to the conventional view, it should be possible to engineer vaccines that provide stronger protection than that provided by natural infection.

## Supporting information

**S1 Text. In S1 Text we describe simulations of the dynamics of infection for insight into the interactions that give rise to the tradeoffs observed when we attenuate various immune-evasion pathways of the virus.** We also present two alternative models for comparison to the basic model of the main text.
(PDF)

## Author Contributions

**Conceptualization:** Rustom Antia, Hasan Ahmed, James J. Bull.

**Formal analysis:** Rustom Antia, Hasan Ahmed, James J. Bull.

**Funding acquisition:** Rustom Antia, James J. Bull.

**Software:** Rustom Antia.

**Visualization:** Rustom Antia, James J. Bull.

**Writing – original draft:** James J. Bull.

**Writing – review & editing:** Rustom Antia, Hasan Ahmed, James J. Bull.

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
