## [Decision Letter · Decision Letter 0]

18 Jun 2020

Dear Prof. Antia,

Thank you very much for submitting your manuscript "Using directed attenuation to enhance vaccine immunity" for consideration at PLOS Computational Biology. As with all papers reviewed by the journal, your manuscript was reviewed by members of the editorial board and by several independent reviewers. The reviewers appreciated the attention to an important topic and found the paper creative and the results useful. Based on the reviews, we are likely to accept this manuscript for publication, providing that you modify the manuscript according to the review recommendations.

In particular, please attend to the comments of reviewer one, who would like further consideration of the generality of these results. 

Sincerely,

Jessica C. Flack

Associate Editor

PLOS Computational Biology

Thomas Leitner

Deputy Editor

PLOS Computational Biology

[LINK]

Reviewer's Responses to Questions

**Comments to the Authors:**

Reviewer #1: This is a well-written paper presenting a novel and interesting idea on the more intelligent design of vaccines. The idea is illustrated by analyzing an elegantly simple model of the innate and adaptive immune system. This modeling is required as results are surprising: in retrospect most parameter combinations that work are intuitive, but I would not have been able to predict these particular combinations beforehand. I also like your figures: very intuitive and clear.

I have only two major recommendations:

1. One can test the generality of the main results, i.e., the 3 parameter combinations giving decreased pathology and increased immunity, by exploring a few alternative models. For instance, a model where the adaptive immune response is (also) limited by antigens.

2. It would be good to see if the main mathematical results can be put into intuitive general conclusions like: prolonging innate immune is beneficial because it extends the period over which the adaptive immune response expands.

Minor comments:

Page 2: Historically, the mechanism of attenuation has been one of genetically reduced viral growth rate,

=> Historically, the mechanism of attenuation has been a reduced viral growth rate, ?

Page 3: (e.g. for mumps where the and necessitates periodic ..

something missing

Page 4: so Z equals the fraction of the the maximum

the the

Page 4: It is justified on biological grounds that the generation of immune responses requires stimulation by innate immunity and that this can occur even after replicating antigen is cleared.

Please provide references for this.

Table 1: Explain the 10^2v .02z: what is v and z?

Page 10: However if an individual was To overcome this problem

lower case T

Page 15: However they (i.e. [57]) suggested

Write our the they

Finally, although I agree that this needs to be worked our for every particular virus, filling in these details and modeling them in an intuitive manner will be a major challenge. Again, it would be good to have a better sense of the generality of the results.

Reviewer #2: This is a very interesting paper, which proposes parameters of the interaction between a pathogen and the immune system which could be targeted in designing a vaccine. The authors suggest that making various changes to these parameters, such as extending the lifespan of innate immunity, could simultaneously lower the pathology of the vaccine and raise immunity to it. Standard vaccines involve a pathogen whose growth rate is diminished. This results in lower pathology, but also lower immunity. The effects of simultanously varying various pairs of parameters of the model of pathogen and immune response are attractively summarised in heat maps of the pathology and immunity. If it is found possible to vary the identified parameters biologically, this could be very useful in designing vaccines.

Table 1 should be referred to in the text and authors should explain how their parameter estimates are derived.

There are also a few tiny errors:

p3 "This can potentially result in reinfection with circulating virus (e.g. for mumps where the and necessitates periodic boosting[12, 17]." This sentence is incomplete.

p4 "The approach to directed attenuation presented here is to develop a model that incorporates key elements of out current understanding of ..." (insert "of")

p10 "However if an individual was to" ("To" -> "to")

p22 "As seen in Fig 3, in both these scenarios.." ("In" -> ", in"; "scenario's" -> "scenarios")

p 22 "As the adaptive immune response is needed for virus clearance we find that the duration of infection is not significantly affected and consequently the amount of adaptive immunity generated does not substantially decrease." (add "decrease")

**Have all data underlying the figures and results presented in the manuscript been provided?**

Reviewer #1: Yes

Reviewer #2: Yes

PLOS authors have the option to publish the peer review history of their article (what does this mean?). If published, this will include your full peer review and any attached files.

Reviewer #1: Yes: Rob de Boer

Reviewer #2: No
---

## [Decision Letter · Decision Letter 1]

2 Dec 2020

Dear Prof. Antia,

We are pleased to inform you that your manuscript 'Directed attenuation to enhance vaccine immunity' has been provisionally accepted for publication in PLOS Computational Biology.

Best regards,

Jessica C. Flack

Associate Editor

PLOS Computational Biology

Thomas Leitner

Deputy Editor

PLOS Computational Biology

Reviewer's Responses to Questions

**Comments to the Authors:**

Reviewer #1: Thank you for carefully responding to my suggestions.

**Have all data underlying the figures and results presented in the manuscript been provided?**

Reviewer #1: Yes

PLOS authors have the option to publish the peer review history of their article (what does this mean?). If published, this will include your full peer review and any attached files.

Reviewer #1: **Yes: **Rob J. de Boer

---

## [Editor Report · Acceptance letter]

26 Jan 2021

PCOMPBIOL-D-20-00748R1 

Directed attenuation to enhance vaccine immunity

Dear Dr Antia,

I am pleased to inform you that your manuscript has been formally accepted for publication in PLOS Computational Biology. Your manuscript is now with our production department and you will be notified of the publication date in due course.

With kind regards,

Alice Ellingham
